# Targeting *SRC* Kinase Signaling in Pancreatic Cancer Stem Cells

**DOI:** 10.3390/ijms21207437

**Published:** 2020-10-09

**Authors:** Sonia Alcalá, Víctor Mayoral-Varo, Laura Ruiz-Cañas, Juan Carlos López-Gil, Christopher Heeschen, Jorge Martín-Pérez, Bruno Sainz

**Affiliations:** 1Department of Cancer Biology, Instituto de Investigaciones Biomédicas “Alberto Sols” (IIBM), CSIC-UAM, 28029 Madrid, Spain; sonia.alcala@uam.es (S.A.); vmayoral@iib.uam.es (V.M.-V.); lauraruiz@iib.uam.es (L.R.-C.); jclopez@iib.uam.es (J.C.L.-G.); jmartin@iib.uam.es (J.M.-P.); 2Department of Biochemistry, Universidad Autónoma de Madrid (UAM), 28029 Madrid, Spain; 3Chronic Diseases and Cancer, Area 3 - Instituto Ramón y Cajal de Investigación Sanitaria (IRYCIS), 28034 Madrid, Spain; 4Stem Cells & Cancer Group, Molecular Pathology Programme, Spanish National Cancer Research Centre (CNIO), 28029 Madrid, Spain; christopher.heeschen@icloud.com; 5Present address: Center for Single-Cell Omics and Key Laboratory of Oncogenes and Related Genes, School of Medicine, Shanghai Jiao Tong University, Shanghai 200240, China

**Keywords:** *SRC* kinases, cancer stem cells, pancreatic ductal adenocarcinoma, PP2, dasatinib, patient-derived xenografts

## Abstract

The proto-oncogene nonreceptor tyrosine-protein kinase *SRC* is a member of the *SRC* family of tyrosine kinases (SFKs), and its activation and overexpression have been shown to play a protumorigenic role in multiple solid cancers, including pancreatic ductal adenocarcinoma (PDAC). PDAC is currently the seventh-leading cause of cancer-related death worldwide, and, by 2030, it is predicted to become the second-leading cause of cancer-related death in the United States. PDAC is characterized by its high lethality (5-year survival of rate of <10%), invasiveness, and chemoresistance, all of which have been shown to be due to the presence of pancreatic cancer stem cells (PaCSCs) within the tumor. Due to the demonstrated overexpression of *SRC* in PDAC, we set out to determine if *SRC* kinases are important for PaCSC biology using pharmacological inhibitors of *SRC* kinases (dasatinib or PP2). Treatment of primary PDAC cultures established from patient-derived xenografts with dasatinib or PP2 reduced the clonogenic, self-renewal, and tumor-initiating capacity of PaCSCs, which we attribute to the downregulation of key signaling factors such as p-FAK, p-ERK1-2, and p-AKT. Therefore, this study not only validates that *SRC* kinases are relevant and biologically important for PaCSCs but also suggests that inhibitors of *SRC* kinases may represent a possible future treatment option for PDAC patients, although further studies are still needed.

## 1. Introduction

It is now generally accepted that the majority of solid tumors contain a small subpopulation of highly plastic, tumorigenic, and chemoresistant cells, known as cancer stem cells (CSCs), that are the drivers of tumor evolution, metastasis, and disease relapse [1,2,3]. Thus, from a clinical perspective, the effective targeting and elimination of CSCs should lead to tumor elimination. Unfortunately, since their discovery in acute myeloid leukemia in 1994 [4], the development of anti-CSC-specific therapies has proven difficult due to the genetic and nongenetic factors underlying the inherent CSC state and the plasticity that exists between CSCs and non-CSCs [5]. As a consequence, to date, only a handful of anti-CSC-specific therapies exist, but many promising therapies are in clinical trials (reviewed in Yang et al. [6]). To target CSCs, researchers have attempted to (1) identify and inhibit pathways or biological properties unique to these cells (reviewed by Yang et al. [6] and Saygin et al. [7]) or (2) induce their differentiation to deplete the pool of self-renewing CSCs. The latter, first proposed by Sachs L. in 1978 [8], would eventually lead to the identification of all-trans-retinoic acid [9] and arsenic trioxide [10] as inducers of cancer cell differentiation and their subsequent combined use in the clinic to treat patients with acute promyelocytic leukemia, resulting in > 93% remission rates and 5-year overall patient survival rates of almost 100% [11]. In 2012, Sachlos et al. showed that thioridazine, a Food and Drug Administration (FDA)-approved antipsychotic dopamine receptor antagonist of the phenothiazine group, could induce differentiation of leukemic stem cells, giving rise to differentiated progeny that enter into normal and terminal cellular life cycles [12]. Thus, targeting CSCs is a realistic goal and warrants research centered on identifying the Achilles’ heel(s) of these cells across different tumor entities.

Pancreatic ductal adenocarcinoma (PDAC) is the most common malignancy of the pancreas. While its incidence is low (age-adjusted annual incidence of 12.9 cases per 100,000 person-years) compared to other cancer types [13], PDAC is one of the deadliest human cancers, with an overall 5-year survival rate of <10% [13]. Currently, PDAC represents the seventh leading cause of cancer-related death worldwide; however, it is predicted to soon become the third leading cause of cancer death in the European Union [14] and the second in the United States [15]. While these alarming statistics can be attributed to the fact that PDAC is typically diagnosed at advanced stages, due to a lack of both symptoms and sensitive/specific markers for early detection [16], the existence of pancreatic CSCs (PaCSCs) [17,18] likely also plays an important role in the poor prognosis of this disease and contributes to the inherent aggressiveness as well as the chemotherapy- and radiotherapy-resistant nature of this tumor. Over the past 13 years, we have come to realize that PaCSCs are not merely a distinct subpopulation of tumor cells, but these cells are biologically different from their non-CSC counterparts on the epigenetic [19], transcriptional [20], metabolic [21], and protein levels [22,23,24,25], and that these differences drive their tumorigenic, chemo-resistant, and metastatic potential, but, at the same time, may represent weaknesses that can be exploited therapeutically.

While the field has unraveled many important signaling nodes and aberrant crosstalk pathways that are important for PaCSC biology [26], we are still far from understanding the biological intricacies of these cells. For example, the proto-oncogene nonreceptor tyrosine-protein kinase *SRC* is overexpressed in PDAC and has been linked to tumor development, progression, and metastasis [27,28,29,30,31,32]; however, the role of *SRC* kinases in PaCSCs has been understudied [33]. The protein *SRC* belongs to the *SRC* family kinases (SFKs), made up of 11 kinase members: *SRC*, FYN (proto-oncogene tyrosine-protein kinase FYN), YES (Yamaguchi sarcoma), BLK (B-lymphoid tyrosine kinase), FRK (FYN-related kinase), YRK (YES-related kinase), FGR (Gardner–Rasheed feline sarcoma), HCK (hematopoietic cell kinase), LCK (lymphocyte-specific kinase), LYN (tyrosine-protein kinase LYN), and SRMS (SRC-related kinase lacking C-terminal regulatory tyrosine and N-terminal myristylation sites) [34,35,36,37]. This family of proteins plays an important role in numerous intracellular signaling pathways involved in proliferation, motility, adhesion, angiogenesis, invasion, and survival. SFKs can respond to extracellular signals, leading to the activation of nuclear factors and their transcriptional programs, or they can interact with cytoplasmic proteins with diverse functions, such as those involved in cytoskeleton reorganization. *SRC*, YES, and FYN, which have been most frequently associated with tumor initiation and metastasis, are the three members of the SFKs that are ubiquitously expressed in all cells [38]. The overexpression, activation, or deregulation of *SRC* and other family members of the SFKs has been associated with different solid tumors (reviewed by Martellucci et al. in [37]), including, for example, ovarian cancer [39,40] and breast cancer [41,42,43]. At the level of CSCs, *SRC* has also been linked to several key CSC-associated pathways or phenotypes, such as EMT, pluripotency, self-renewal, and metastasis [44,45,46,47,48,49,50], but, as mentioned above, the role of *SRC* kinases in PaCSCs has been underexplored. Thus, in this study, we sought to determine whether *SRC*-kinases are biologically important for PaCSCs using primary PDAC cultures established from patient-derived xenografts (PDXs) that we have previously shown to contain PaCSCs [25]. Using pharmacological inhibitors of *SRC* kinases, we show that *SRC* kinases and *SRC*-related signaling is biologically important for PaCSCs at the level of gene expression, self-renewal, and tumor-initiating capacity. Therefore, inhibitors of *SRC* kinases can certainly affect PaCSCs and the data presented herein support the continued evaluation of these inhibitors as possible treatment options, likely in combination with standard of care, for PDAC.

## 2. Results

### 2.1. *SRC* is Overexpressed in PDAC

*SRC* has been shown by others to be frequently activated and overexpressed in PDAC [28,29,30,31,32]. We verified these findings in publicly available transcriptome PDAC datasets that have become available over the past 10 years (META dataset [51], Jandaghi et al. [52], The Cancer Genome Atlas (TCGA; http://xena.ucsc.edu), and Janky et al. [53]). The transcriptional levels of *SRC* expression were evaluated in normal and tumor samples, and we observed that *SRC* mRNA expression is significantly elevated in whole pancreatic tumor samples versus adjacent normal tissue for the META and Jandaghi et al. datasets (Figure 1A). For The Cancer Genome Atlas (TCGA) dataset, we used the curated dataset by Nicolle et al. [54] to separate samples into nontumor (i.e., normal pancreas, atrophic pancreas or noninvasive intraductal papillary mucinous neoplasms), neuroendocrine tumors (NET), or PDAC, and again observed higher levels of *SRC* in PDAC versus nontumor samples (Figure 1B). In the study by Janky et al., samples were classified as normal pancreas, early or advanced PDAC, and lymph node metastasis (LNM). *SRC* levels were significantly higher in early and advanced samples versus adjacent normal samples, and an even more significant increase was observed in LNM versus adjacent normal samples (Figure 1C). It has been recently shown that PDAC tumors can be classified into two main subtypes: (1) classical or progenitor versus (2) basal, squamous, or quasi-mesenchymal [55,56,57]. Depending on the gene signatures used by the authors to subtype the tumors, differences do exist; however, it is well accepted that the basal, squamous, or quasi-mesenchymal subtypes are associated with a more aggressive phenotype with lower overall survival but, at the same time, are more sensitive to gemcitabine and/or benefit from adjuvant chemotherapy [58]. Using the Bailey et al. subtype profiles [56], we observed that *SRC* expression was less associated with the squamous subtype, as well as the aberrantly differentiated endocrine exocrine (ADEX) subtype, and expressed to higher levels in the progenitor subtype (Figure 1D).

Next, we took advantage of the well-annotated clinical data available for the TCGA dataset to determine if high levels of *SRC* expression correlated with decreased overall survival. No clear deviation or significant decrease in the median overall survival of *SRC* high-expressing patients (the top 25%) compared to the *SRC* low-expressing patients (bottom 25%) was observed in the TCGA dataset (Figure 1E), suggesting that *SRC* overexpression is not a marker of patient survival. Finally, we performed gene set enrichment analysis (GSEA), comparing the patient samples expressing high levels of *SRC* to those expressing low levels of *SRC* from the TCGA, META [51], Jandaghi et al. [52], and Bailey et al. [56] datasets (Figure 1F and Appendix A). While *SRC* expression could not stratify patients at the level of overall survival, at the level of GSEA, we observed significantly and commonly enriched pathways linked to *SRC* expression across all four series analyzed. Specifically, using the Hallmark genesets collection, pathways such as mTOR, NOTCH, Wnt/β-catenin, TGF-ß, and MYC signaling, as well as metabolic pathways, including glycolysis and oxidative phosphorylation, were enriched in patient samples expressing high levels of *SRC* (Figure 1F,G and Appendix A).

### 2.2. *SRC* Kinases are Overexpressed in PaCSCs

Since many pathways involved in CSC biology were enriched in the *SRC* high population, we questioned whether *SRC* kinases had a biological role in PaCSC biology. Towards this end, we first took advantage of published RNA sequencing (RNAseq) data (ArrayExpress: E-MTAB-3808 [21]) to confirm whether CSC-enriched cultures overexpress *SRC*. Specifically, RNAseq was performed on adherent- and sphere-derived cultures from a panel of PDX PDAC primary cultures. We observed that *SRC* mRNA levels were significantly higher in CSC-enriched sphere cultures compared to their adherent counterparts (Figure 2A). We validated these findings at the protein level in lysates from adherent- and sphere-derived cultures from 6 PDX-derived cultures and observed increased *SRC* kinases and pY419–*SRC* protein levels in spheres in 5 out of the 6 cultures analyzed (Figure 2B). Together, these data suggested that *SRC* kinases are overexpressed in PaCSCs. 

### 2.3. Inhibition of *SRC* Kinases Affects Several Key PaCSC-Associated Signaling Factors

To assess the biological relevance of *SRC* kinases in PaCSCs, we initiated pharmacological inhibition-based experiments using two broad-specificity tyrosine kinase inhibitors, dasatinib (DAS) and PP2 [42], in three selected PDX-derived cultures: Panc185, Panc354, and Panc253. Pyrazolopyrimidine compound PP2 is primarily known as an inhibitor of SFKs [59], but PP2 can also inhibit other kinases, such as the kinase activity of the TGF-β type I receptor activin receptor-like kinase 5 [60]. DAS (BMS-354825, Sprycel) is a potent small molecule that inhibits more than 40 different protein kinases, including SFKs and other non-RTKs such as FRK, BRK, and ACK [61]. DAS is orally available and FDA-approved to treat chronic myelogenous leukemia and Philadelphia-positive acute lymphoblastic leukemia [62]. For the inhibition experiments presented herein, PP2 was used at a concentration of 5 µM and DAS at a concentration of 100 nM, based on their relatively low toxic effects (Appendix A).

First, we performed an analysis of the expression of SFKs and downstream signaling factors associated with SFKs to assess their expression in PaCSCs and determine the intracellular consequences of their inhibition using DAS and PP2. For these assays, PaCSCs were enriched for by culturing PDX-derived cells in anchorage-independent conditions, and treatments were added on d0 and d5 post-sphere formation. Spheres were harvested on d7 post-seeding. Western blotting (WB) assays were performed on cell extracts using a polyclonal antibody (*SRC* kinases) that recognizes the C-terminal portion common to *SRC*, YES, and FYN (62 to 59 MW range), and a pY419–*SRC* polyclonal antibody, which identifies the highly conserved autophosphorylation site within the catalytic domain of *SRC* kinases [63]. Autophosphorylation enhances the enzymatic activity of SFKs [64,65]. Treatment of Panc185 and Panc253 PaCSCs with either DAS (100 nM) or PP2 (5 µM) reduced the ratio of pY419–SRC/*SRC* kinases, indicating that DAS and PP2 effectively reduce the activity of *SRC* kinases in PaCSC cultures (Figure 3A). It is well established that *SRC* binds throughout its SH2 domain to pY379–FAK (the FAK autophosphorylated site), causing *SRC* open conformation and activation. In turn, activated *SRC* phosphorylates FAK at Y-925, which serves as a docking site for SH2-domain-containing proteins such as GRB2 [66], facilitating intracellular signaling. On this basis, we tested for the effect of DAS and PP2 treatment on *SRC* phosphorylation of pY925–FAK. The ratio of pY925-FAK/FAK was clearly inhibited in Panc185 and Panc253 PaCSCs, indicating that the tyrosine kinase activity of *SRC* kinases was blocked by DAS and PP2 (Figure 3A). We additionally confirmed the ratio of pY419–SRC/*SRC* kinases and the ratio of pY925–FAK/FAK in Panc354 cells and observed a similar inhibitory effect upon DAS and PP2 treatment (Appendix A).

The aforementioned data suggested that *SRC* kinase activity is active in PaCSCs (Figure 3A); therefore, we analyzed the capacity of the inhibitors to affect cell-proliferation-associated proteins associated with SFKs in Panc185 and Panc253 PaCSCs. Both DAS and PP2 increased expression of the G1-phase cell cycle inhibitor p27^kip1^ in Panc185 and Panc253 PaCSCs (Figure 3B). Consistently, the inhibitors reduced the expression of cyclin D1 and MYC (Figure 3B). Furthermore, the activation of ERK1-2 by MEK1-2 phosphorylation at T202/Y204 was also reduced upon treatment of cells with either inhibitor (Figure 3B). Together, these results indicate that *SRC* kinases are clearly involved in pathways controlling PaCSC proliferation. There is evidence that *SRC* tyrosine phosphorylates AKT [67], and, via the canonical pathway, *SRC*/FAK/GRAB2-SOS/PI3K induces its phosphorylation at T308 and S473 [68,69], facilitating its activation. AKT stimulation has several effects: (1) it phosphorylates the cell cycle inhibitor p27^kip1^, impeding its nuclear translocation; (2) it phosphorylates GSK3 at S9, inhibiting its activity; (3) it phosphorylates FOXO3A at T32 and promotes its nuclear exclusion. Through these combined actions, AKT promotes MYC expression and blocks its degradation, which, in turn, facilitates proliferation [69,70]. Here, we observed that the ratio pS473–AKT/AKT was slightly reduced by incubating Panc185 and Panc253 PaCSCs with DAS, while PP2 only reduced AKT activation in Panc185 CSCs (Figure 3B). MYC expression was clearly reduced in Panc253 by both DAS and PP2 treatment. In contrast, cyclin D1 was mainly inhibited by PP2 in Panc253 (Figure 3B). Similarly, Panc253 was more sensitive than Panc185 to the effects of DAS and PP2 in inhibiting ERK1-2 activation (Figure 3B). Together, these results support the requirement of the activity of *SRC* kinases in the biology of PaCSCs.

### 2.4. Inhibition of *SRC* Kinases Reduces the Percentage of PaCSCs, Pluripotency-Associated Gene Expression, and Colony Formation Efficiency

To investigate the consequences of inhibition of *SRC* kinases in PaCSCs, we next evaluated the effect of PP2 or DAS on key PaCSC phenotypic and functional properties. First, to assess the effect of inhibiting *SRC* kinases on the PaCSC population, Panc185 cells were treated for 24 h with PP2 or DAS, and the percentage of CD133-positive cells was determined by flow cytometry 48 h later. A marked reduction in the percentage of CD133-positive cells was observed (Figure 4A), which, when quantified and extended to a larger panel of PDX-derived cells, showed a significant reduction in all three cultures tested (Figure 4B). Similar results were observed with CXCR4 and with the intracellular PaCSC marker autofluorescence [25], which is the result of riboflavin accumulation in ABCG2-coated intracellular vesicles present in PaCSCs (Figure 4B). Importantly, the effect was similar, regardless of the inhibitor used. Together, these data indicated that inhibition of *SRC* kinases reduces the CSC population in PDX-derived PDAC cells.

To confirm the latter, pluripotency-associated genes were analyzed in PP2- or dasatinib-treated cultures. The Yamanaka factors [71], including *KLF4*, *POU5F1* (OCT3/4), and *SOX2*, are generally overexpressed in PaCSCs [72]. While essentially no reduction was observed in Panc185 cells, *KLF4*, *POU5F1*, and *SOX2* levels were significantly reduced in Panc253 and Panc354 cells following PP2 or DAS treatment (Figure 5A), indicating that inhibition of *SRC* kinases affects transcriptional properties associated with PaCSCs or the PaCSC pool is eliminated by PP2 or DAS treatment. We hypothesized that the aforementioned results were a consequence of a reduction in the CSC population in the tested PDX-derived cultures. Thus, we tested the clonogenic capacity of PP2- or DAS-treated cultures in a standard colony formation assay lasting 11 days to functionally determine the remaining PaCSC pool, following PP2 or DAS treatment. To ensure that the inhibition of *SRC* kinases was maintained throughout the course of the assay, parallel cultures were retreated with PP2 or DAS on Day 7 post-seeding. In line with our flow cytometric and RTqPCR analyses, inhibition of *SRC* kinases significantly reduced the clonogenic capacity of PaCSCs across all three PDX-derived cultures, independent of the inhibitor used, and the effect was more pronounced when treatments were replenished on Day 7 (Figure 5B).

### 2.5. *SRC* Kinase Inhibition Reduces PaCSC Self-Renewal and Tumorigenesis

Finally, to evaluate the functional consequence of the inhibition of *SRC* kinases on PaCSCs, we assessed the self-renewal and tumor initiation capacity of PaCSCs in vitro and in vivo, respectively. To assess self-renewal, PDX-derived cells were cultured in anchorage-independent conditions, and treatments were added at the initiation of each generation of spheres (d0 (first generation), d7 (second generation), and d14 (third generation)). To ensure that the inhibition of *SRC* kinases was maintained throughout the course of the assay, parallel cultures were retreated with the inhibitors on the fifth day of each generation of spheres (d5 (first generation), d12 (second generation), and d19 (third generation)). Consistent with the effects observed in the colony formation assays and on key CSC-associated signaling factors, PaCSC self-renewal was significantly reduced in Panc253 and Panc354 cells across three generations, with a more potent effect observed with PP2 and when cells were retreated (Figure 6A). Interestingly, although the effect of single-treatment administration of PP2 or DAS was initially less potent, by the third generation, PaCSCs self-renewal was reduced to levels equivalent to that achieved in the retreated groups, suggesting that inhibition of *SRC* kinases is long-lasting and significantly affects PaCSCs.

To translate these findings to the in vivo setting, we performed an extreme limiting dilution assay (ELDA) with Panc354 cells pretreated for 24 h with 5 µM PP2. PP2 was chosen over DAS due to the inappreciable toxicity associated with PP2 (Appendix A). Ten weeks post-injection, PP2-pretreated cells formed fewer tumors, a phenotype that was particularly evident when low numbers of cells (i.e., 500 cells) were injected (Figure 6B). The tumors that did form, however, were similar in weight between treated and control-diluent-treated groups, although a notable but nonsignificant trend towards smaller tumors was observed for tumors derived from 500 injected PP2-pretreated cells (Figure 6C). Lastly, we calculated the CSC frequency using ELDA software. A CSC frequency of 1 in 3210 was calculated for control-diluent-treated cells, while in PP2 pretreated cells, the CSC frequency was significantly reduced by approximately 3-fold (1 in 9032 cells; Figure 6D).

## 3. Discussion

*SRC* mutations have not been consistently defined across human tumors (TCGA; http://xena.ucsc.edu). Nevertheless, overexpression and/or hyperactivation of *SRC* occurs in a variety of solid tumors, including breast, prostatic, colorectal, and pancreatic tumors [37,47,65,73]. In PDAC, *SRC* kinase activity has been shown to be required for proliferation, migration, invasion, anchorage-independent growth, and survival, as treatment with DAS or small interfering RNAs to *SRC* inhibited all of these biological processes [74]. In addition, DAS treatment also inhibited in vivo pancreatic tumor growth [74]. Similarly, RNA-interference-based knockdown of SFKs significantly inhibited proliferation, migration, and invasiveness of pancreatic cancer cells [31]. However, as described above, targeting the small population of CSCs should be the aim of therapeutic treatments, as these are the cells responsible for tumor relapse and chemoresistance [1,2,3]. In this context, it has been recently shown that *SRC* functionality is relevant for MCF7 breast CSC proliferation and self-renewal by, at least in part, regulation of CSC glucose metabolism [75]. Thus, while *SRC* has been well explored and its role characterized in CSCs of other tumor entities [44,45,46,47,48,49,50], little is known regarding the role of *SRC* in PaCSCs and whether targeting *SRC* kinases in PaCSCs is therapeutically relevant.

Here, we have described that *SRC* is upregulated in PaCSCs and explored the role of the activity of *SRC* kinases in sphere-enriched populations of PaCSCs derived from several PDAC PDXs. The increase in *SRC* kinases in PaCSCs is not surprising as *SRC* expression and activity is linked to the ligand/receptor signaling pathways commonly overexpressed in CSCs, such as EGF/EGFR, PDGFR, IGF-1R, HGF/c-MET, c-KIT, VEGF/VEGFR, FGFR, and IL-6/IL-6R-gp130, among others [76,77,78]. Thus, overexpression of *SRC* kinases is likely necessary for the maintenance of the PaCSC-state. Indeed, treatment of Panc185, -253, or -354 PDX-derived CSCs, with either PP2 or DAS, resulted in inhibition of cellular CSC-associated properties, such as CSC cell surface marker expression, pluripotency-associated gene expression, colony formation efficiency, self-renewal, and tumorigenicity, using a limiting dilution assay approach that facilitates the calculation of the frequency of CSCs. The most profound effects in vitro were observed at the level of colony and self-renewal efficiency, which were reduced upon treatment with PP2 or DAS. Interestingly, we did observe a difference across the three PDX-derived cultures tested at the level of response kinetics, with Panc185 cells exhibiting less sensitivity to DAS and PP2 compared to Panc253 and Panc354 cells. This could indicate that while *SRC* kinases are important for PaCSCs, the dependence of PaCSCs on *SRC* kinases may differ from patient to patient. The latter may be PDAC-subtype-specific; however, this study did not consider subtype differences between Panc185, -253, or -354 PDX-derived CSCs. Using the Bailey et al. dataset to determine *SRC* (*SRC)* levels across the 4 PDAC-subtype profiles described by these authors [56], we did observe that *SRC* is expressed to higher levels in the progenitor subtype compared to the squamous subtype. This might suggest that *SRC* may have a more important role in progenitor or classical tumors, which have been associated with higher resistance to gemcitabine/adjuvant chemotherapy [58], and, thus, these tumors may benefit from anti-*SRC* therapy. However, a recent phase I study using gemcitabine plus DAS or gemcitabine plus DAS plus cetuximab in refractory solid tumors showed that the clinical efficacy of gemcitabine plus DAS was modest, and the authors concluded that the results did not support further investigation of this combination treatment approach for PDAC [79]. PDAC subtypes were not taken into consideration in this study, and while it is uncertain whether such stratification would have impacted the results, based on the aforementioned link between *SRC* RNA levels and the progenitor subtype, it may be worth exploring the level of *SRC* (and other SFKs) at the protein levels in a large cohort of PDAC patients to confirm the correlation observed herein and to determine whether *SRC* may have a prognostic value.

*SRC* acts as a “switcher”, responding to very different extracellular signals generated by the activation of cytokine and growth factor receptors, integrins, and so forth. The signal transduction pathways regulated by *SRC* control numerous important biological processes, and, depending on the cellular context, *SRC* can modulate proliferation, survival, differentiation, migration, and invasion. Evidence from several experimental model systems supports our findings linking *SRC* kinases with important PaCSC phenotypes. For example, in triple-negative breast cancer stem cells (TN-BCSCs) both *SRC* kinase activity and BCSC sphere formation efficiency were highly elevated in paclitaxel-resistant cells [49]. DAS inhibition of *SRC* kinase activity resulted in the abrogation of TN-BCSC self-renewal induced by paclitaxel, indicating that resistance to paclitaxel and BCSC expansion is regulated by *SRC* kinases [49]. In 5-fluorouracil (5FU)-resistant human pancreatic cells, inhibition of the activity of *SRC* kinases by PP2 restored cellular sensitivity to 5-FU, and combined treatment of 5FU-resistant pancreatic cells with 5-FU plus PP2 inhibited 5-FU activation of the EGFR/SRC/AKT pathway as well as tumor growth and distant metastasis in vivo [80]. Additionally, in pancreatic tumor cells, DAS blocked TGFß1-induction of SMAD phosphorylation, migration, and epithelial/mesenchymal-associated-gene expression and invasion. In addition, in a study by Bartscht et al., the authors showed that DAS inhibited gene expression associated with PaCSCs as well as single-cell colony formation capacity, which, in turn, prevented the metastatic spreading of pancreatic ductal carcinomas [81]. CD133, a well-established CSC maker [18], interacts with *SRC* at its C-terminal portion, which, in turn, activates the *SRC*–FAK complex and the phosphorylation of FAK at Y925, facilitating cell migration in the colorectal cancer cell line SW620 through the CD133/SRC/FAK pathway, which could be blocked by PP2 [82]. In NSCLC cells, side population cells express elevated levels of the stem cell markers OCT4, NANOG, and SOX2 under the control of EGFR/SRC/AKT. The authors show that DAS and PP2 significantly reduced SOX2, p-AKT, p-ERK1-2, p-EGFR, and p-SRC, blocking CSC self-renewal capacity [83]. Finally, in MCF7, SUM159PT and MDA-MB-231 breast cancer cells, multiple groups have shown that *SRC* controls pathways that regulate proliferation, survival, migration, invasion, and stemness [65,75,84]. Similarly, in this study, we show that in PaCSC-enriched cultures, inhibitors of *SRC* kinases modulate key signaling pathways linked to proliferation and survival, resulting in reduced sphere formation and colony formation efficiency as well as a clear reduction of the tumorigenic properties of these PDAC PaCSCs in vivo. Of the targets consistently reduced by DAS and PP2 treatment was p-FAK (Y925). FAK is a nonreceptor tyrosine kinase, which mediates signals transmitted by growth factor receptors and integrins. FAK and *SRC* can physically and functionally interact to promote downstream signaling [85], and FAK activation has been associated with CSCs in a number of tumors. For example, Kolev et al. showed that administration of VS-4718 or VS-6063, two potent FAK inhibitors, to mice bearing TNBC xenografts significantly reduced the CSC population [86]. Recently, it was shown in PDAC that FAK activation promotes PaCSCs, and short hairpin RNAs against *FAK* or a small-molecule FAK inhibitor can inhibit PaCSC clonogenic growth [87]. Thus, the inhibition observed in the PaCSC population in this study with DAS and PP2 may very well be due to the *SRC*/FAK axis. We also observed a reduction in p-ERK1-2 and p-AKT, downstream *SRC* effector molecules that we have previously shown to be important for PaCSCs [88,89]. Reduction in p-ERK1-2 or p-AKT may also have affected PaCSC properties, resulting in the observed phenotypes. It is important to stress that PP2 and DAS are broad-specificity tyrosine-kinase inhibitors, as mentioned above. Thus, while both compounds can target and reduce the activity of SFKs, as demonstrated herein, their effects may reach beyond that of SFKs. Therefore, we cannot rule out the possibility that the biological and functional effects observed with PP2 and DAS may be due to SFK-independent effects. As such, genetic-based approaches to silence or knockout *SRC* and/or other SKFs, specifically in PaCSCs, are warranted.

In conclusion, agents that target SFKs or FAK are clinically available and/or are under development; however, as mentioned above, inhibitors of *SRC* kinases (e.g., DAS) have been shown to be ineffective in PDAC patients, even when combined with gemcitabine as shown in the Mettu et al. study [79]. It should be noted that the gatekeeper threonine (T) at *SRC* amino acid 341 is related to DAS resistance, as a T341I–*SRC* variant made different cell types insensitive to *SRC* tyrosine kinase inhibition by DAS [90,91,92]. It would be interesting to assess the status of T341 in the PDAC patients treated in the Mettu et al. study and in the PDX-derived cultures used in the study, particularly Panc185. Nonetheless, resistance to targeted therapies, in general, is very common in PDAC and PaCSCs. For example, MEK inhibitors are effective against PaCSCs [88], but PDAC tumors quickly adapt [93]. Thus, combinatorial approaches incorporating inhibitors of *SRC* kinases together with mutational profiling will likely be more effective and advantageous for the treatment of PDAC. For example, the *SRC*/FAK/JAK2 (Janus kinase 2) inhibitor TPX-0005 was used in combination with gefitinib or osimertinib and showed potent inhibition in NSCLC xenografts in vivo [85]. Thus, inhibition of *SRC* kinases, in combination with inhibitors targeting other key PaCSC effectors, may represent the future of PDAC therapy.

## 4. Materials and Method

### 4.1. Gene Expression Datasets, Gene Set Enrichment Analysis (GSEA), and Kaplan–Meier Analyses

The gene expression datasets used in this study are publicly available. The dataset from Janky et al. [53] was downloaded from GEO (GSE62165); the dataset from Jandaghi et al. [52] was downloaded from ArrayExpress (E-MTAB-1791); the dataset from Bailey et al. was included in a Appendix A of their published work [56]; the META dataset, containing datasets GSE15471, GSE16515, GSE22780, and GSE32688, was generated as described in [51]; the TCGA dataset was downloaded from The Cancer Genome Atlas (TCGA; http://xena.ucsc.edu). The samples included in the top and bottom quartile of expression of *SRC* were compared in GSEA using the Hallmark genesets database. The GSEA module of the Genepattern suite (v4.0.3) from the Broad Institute (MIT, Cambridge, MA, USA) was used, with 1000 permutations, and FDR <25% was considered statistically significant.

To analyze the prognostic value of *SRC* mRNA expression, TCGA patients were stratified into quartiles (top and bottom 25%), and survival analysis was performed with R. The two patient populations were compared by a Kaplan–Meier survival plot, and the hazard ratio with 95% confidence intervals and log-rank *p*-value were calculated. The Cox proportional hazard regression model was used to calculate the hazard ratio. 

### 4.2. Primary Human Pancreatic Cancer Cells and Reagents

The PDAC patient-derived xenograft (PDX) Panc185, Panc253, and Panc354 were obtained from Dr. Manuel Hidalgo under a material transfer agreement with the Spanish National Cancer Centre (CNIO), Madrid, Spain (Reference no. I409181220BSMH). Primary PDX-derived in-vitro cultures were established as previously detailed [25]. Briefly, xenografts were minced, enzymatically digested with collagenase (Stem Cell Technologies, Vancouver, Canada) for 60 min at 37 °C and, after centrifugation for 5 min at 1800 rpm, the cell pellets were resuspended and cultured in RPMI 1640 (1:1; #61870-010; Gibco, Waltham, MA, USA) supplemented with 10% FBS (Invitrogen, Waltham, MA, USA), penicillin/streptomycin ((Pen/Strep; 1:100; #1500-063; Gibco, Waltham, MA, USA), and fungizone (1:250; #15290-018; Gibco, Waltham, MA, USA). Primary cultures were tested for mycoplasma at least every 4 weeks. 

PP2 (Tocris, Ellsville, MN, USA) and dasatinib (LC-Laboratories, Woburn, MA, USA) were resuspended in DMSO and used at 5 µM and 100 nM, respectively, for all the experiments presented herein. Treatments were added as indicated. 

### 4.3. Sphere Formation Assay 

Pancreatic CSC spheres were generated as previously described [25] in ultra-low attachment plates (Corning, Corning, NY, USA) using CSC media: serum-free Dulbecco’s modified Eagle medium (DMEM)/F12 medium (1:1; #21331-020; Gibco, Waltham, MA, USA), supplemented with Pen/Strep (1:100; #1500-063; Gibco, Waltham, MA, USA), fungizone (1:250; #15290-018; Gibco), L-glutamine (1:100; #25030-024; Gibco, Waltham, MA, USA), B-27 (1:50; #17504-044; Gibco, Waltham, MA, USA), and basic fibroblast growth factor (FGF-b; 1:5000; Gibco, Waltham, MA, USA). All cell culture was carried out at 37 °C in a 5% CO_2_ humidified incubator. To quantify spheres, 1 mL of sample volume was analyzed with an inverted EVOS FL microscope (Thermo Fisher Scientific, Waltham, MA, USA) using a 10X objective with phase contrast. For serial passaging, spheres were harvested using a 40-µm cell strainer, trypsinized into single cells, and recultured for an additional 7 days for each generation. Sphere counts are represented as the fold change in sphere numbers/mL.

### 4.4. Flow Cytometry

Cells were analyzed with a 4-laser Attune NxT acoustic cytometer (ThermoFisher Scientific, Waltham, MA, USA). Cells and digested tumors (as described in [22]) were resuspended in FLOW buffer (1X PBS; 3 mM EDTA (*v/v*); 3% FBS (*v/v*)), and the following fluorescent-tagged antibodies were used to label cells for 30 min at 4 °C: mouse monoclonal antihuman CD133-APC (1:20, Cat no. 130-111-080, Miltenyi, Bergisch Gladbach, Germany) or mouse monoclonal antihuman C-X-C chemokine receptor type 4-phycoerythrin (CXCR4-PE; 1:20, Cat no. 130-117-354, Miltenyi, Bergisch Gladbach, Germany). Autofluorescent cells were excited with a blue laser 488 nm and selected as the intersection with the filters 530/40 and 580/30, as previously described [25]. For all assays, DAPI (4′,6-Diamidine-2′-phenylindole dihydrochloride, Cat no. 10236276001, Sigma, St. Louis, MO, USA) was used to mark and exclude dead cells, and data were analyzed using the software FlowJo v9.3 (Tree Star Inc., Ashland, OR, USA).

### 4.5. RNA Preparation and Real-Time qPCR

Total RNA was isolated by the guanidine thiocyanate method using standard protocols [94]. One µg of purified RNA was used for cDNA synthesis using the Thermo Scientific Maxima First Strand cDNA Synthesis Kit (Cat no. K1672, ThermoFisher Scientific, Waltham, MA, USA), followed by SYBR green RTqPCR using an Applied Biosystems 7500 real-time thermocycler (Applied Biosystems, Waltham, MA, USA). Thermal cycling consisted of an initial 10 min denaturation step at 95 °C, followed by 40 cycles of denaturation (15 sec at 95 °C) and annealing/extension (1 min at 60 °C). mRNA copy numbers were determined relative to standard curves comprised of serial dilutions of plasmids containing the human *KLF4*, *SOX2*, or *POU5F1* coding sequences, respectively, and normalized to ß-actin levels. Primers sequences are as follows: human *KLF4* Forward 5′-ACCCACACAGGTGAGAAACC-3′ and Reverse 5′-ATGTGTAAGGCGAGGTGGTC-3′; human *SOX2* Forward 5′-AGAACCCCAAGATGCACAAC-3′ and Reverse 5′-CGGGGCCGGTATTTATAATC-3′; human *POU5F1* (Oct3/4) Forward 5′-CTTGCTGCAGAAGTGGGTGGAGGAA-3′ and Reverse 5′-CTGCAGTGTGGGTTTCGGGCA-3′; human (*ACTB*) ß-actin Forward 5′-GCGAGCACAGAGCCTCGCCTT-3′ and Reverse 5′-CATCATCCATGGTGAGCTGGCGG-3′. 

### 4.6. Colony Formation Assays

For colony formation assays, 750–1000 cells were plated in six-well plates and grown for 11 days in complete culture media. Colonies were fixed in 10% formalin (Cat no. 15812-7, Sigma, St. Louis, MO, USA), stained with 0.025% aqueous solution of crystal violet (Cat no. C3886; Sigma, St. Louis, MO, USA) for 1 h at room temperature on a shaker, washed with distilled water, and, subsequently, lysed for 4 h with 1X PBS/1% SDS (Cat no. L4509, Sigma, St. Louis, MO, USA). Then, 100 µL of lysates was transferred to a U-bottom 96-well plate in triplicate, and optical densities (O.D.) of the solutions were determined at 562 nm using a Synergy HT microplate reader (BioTek, Winooski, VT, USA); results are represented as arbitrary units.

### 4.7. In Vivo Assays

Female 6- to 8-week-old NU-Foxn1nu nude mice (Envigo, Spain) were subcutaneously injected with 10,000, 1000, or 500 control or PP2-pretreated PDAC Panc354 cells in 50 µL of Matrigel^TM^ (Cat no. 734-0270, Corning, Corning, NY, USA). Tumor growth was monitored biweekly for up to 10 weeks. If a mouse in a specific dilution group warranted sacrifice (e.g., ulcerated tumor), all of the mice (Control and Treated) in that dilution group were sacrificed in order to obtain the total number of tumors for all mice at the exact same time. Mice were sacrificed at 4, 8, and 10 weeks post-inoculation for the 10,000-, 1000-, and 500-cell dilutions, respectively.

Mice were housed according to institutional guidelines, and all experiments were performed in compliance with the institutional guidelines for the welfare of experimental animals approved by the Universidad Autónoma de Madrid Ethics Committee (CEI 60-1057-A068) or by the Instituto de Salud Carlos III Ethics Committee (CBA12_2014-v3) and La Comunidad de Madrid (PROEX 335/14 or PROEX 53/14) and in accordance with the guidelines for Ethical Conduct in the Care and Use of Animals, as stated in The International Guiding Principles for Biomedical Research involving Animals, developed by the Council for International Organizations of Medical Sciences (CIOMS).

### 4.8. Western Blotting Assays

Preparation of cell lysates from cells was carried out as previously described [75]. Briefly, cells were washed twice with cool PBS and lysed at 4 °C with lysis buffer (10 mM Tris–HCl (pH 7.6), 50 mM NaCl, 30 mM sodium pyrophosphate, 5 mM EDTA, 5 mM EGTA, 0.1% SDS, 1% Triton X-100, 50 mM NaF, 0.1 mM Na3VO4, 1 mM PMSF, 1 mM benzamidine, 1 mM iodoacetamide, and 1 mM phenantroline). Cell lysates were obtained by centrifugation at 21,380× *g* for 30 min at 4 °C; protein concentration in the supernatant was determined by BCA protein assay (Cat no. 23227, Pierce, Rockford, IL, USA), and lysates were adjusted to equivalent concentrations with lysis buffer. Aliquots of 10–40 µg of total cell lysate were then separated on SDS–PAGE. Proteins were transferred to PVDF membranes (Amersham™ Hybond^®^ P Western blotting membranes, PVDF, Cta no. GE10600021, Sigma, St. Louis, MO, USA) that were blocked overnight at 4 °C with 5% nonfat milk in TTBS (TBS with 0.05% Tween-20). Incubation was first was carried out overnight with primary specific antibodies (Appendix A) at 4 °C and then with horseradish peroxidase-conjugated secondary antibodies in blocking solution for 1 h at room temperature. Immunoreactive bands were visualized by ECL using a MyECL^TM^ Imager (ThermoFisher Scientific, Waltham, MA, USA). Uncropped and unmodified images of immunoblots are provided in Appendix A. 

### 4.9. RNA Sequencing

Illumina RNA sequencing of five primary PDX-derived PDAC cultures grown as either adherent (non-CSCs) or anchorage-independent spheres (CSCs), in duplicate, has been previously performed [21], and the raw data were deposited in ArrayExpress accession number E-MTAB-3808. Differential expression of genes across the different conditions was calculated with Cuffdiff.

### 4.10. Statistical Analyses

Results are presented as means ± standard error of the mean (SEM) unless stated otherwise. Pair-wise multiple comparisons were performed with one-way ANOVA with Dunnett’s test. Unless otherwise stated, unpaired two-sided (confidence interval of 95%) Student’s *t*-tests were used to determine differences between the means of two groups. *p*-values < 0.05 were considered statistically significant. All analyses were performed using GraphPad Prism version 6.0 (San Diego, CA, USA). CSC frequencies and significance were determined using the online extreme-limiting dilution assay (ELDA) program (http://bioinf.wehi.edu.au/software/elda/) [95].

### 4.11. Data Availability

Unique identifiers for publicly available datasets are indicated.

## Figures and Tables

**Figure 1 ijms-21-07437-f001:**
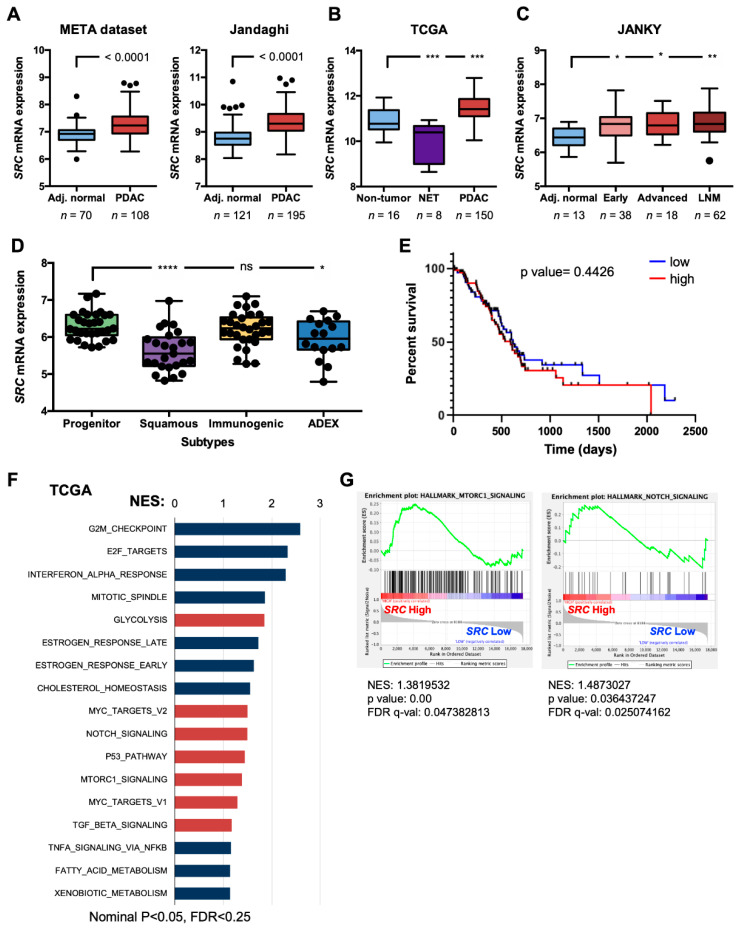
*SRC* is overexpressed in pancreatic ductal adenocarcinoma (PDAC). (**A**) Box and whisker Tukey plots showing the differential expression of *SRC* in normal adjacent (Adj.) tissue versus PDAC tumors in 2 independent transcriptomic data series: META dataset (70 Adj. normal, 108 tumors), Jandaghi et al. (E-MTAB-1791; 121 Adj. normal, 195 tumors). The two whiskers indicate the minimum and maximum values using the Tukey method, and outliers are depicted as black-filled circles (unpaired two-sided Student’s *t*-test). (**B**) Box and whisker Tukey plots showing the differential expression of *SRC* in nontumor versus neuroendocrine tumors (NET) versus PDAC tumors in The Cancer Genome Atlas (TCGA) pancreatic adenocarcinoma (PAAD) dataset. The number of samples (*n*) per group is shown below. The two whiskers indicate the minimum and maximum values using the Tukey method, and outliers are depicted as black-filled circles (*** *p* < 0.001, as determined by one-way ANOVA with Dunnett post-test, comparing NET and PDAC to nontumor). (**C**) Box and whisker Tukey plots showing the differential expression of *SRC* in Adj. normal versus PDAC tumors, classified as Early PDAC, Advanced PDAC, or Lymph Node Metastasis from the Janky et al. dataset (GSE62165). The number of samples (*n*) per group is shown below. The two whiskers indicate the minimum and maximum values using the Tukey method, and outliers are depicted as black-filled circles (* *p* < 0.05, ** *p* <0.01, as determined by one-way ANOVA with Dunnett post-test, comparing Early, Advanced, and LNM to Adj. normal). (**D**) Box and whisker plots showing the differential expression of *SRC* in PDAC tumors, subtyped as Progenitor, Squamous, Immunogenic, or ADEX from the Bailey et al. dataset. The two whiskers indicate the minimum and maximum values, and all points are depicted as black-filled circles (* *p* < 0.05, **** *p* < 0.0001, or ns = not significant, as determined by one-way ANOVA with Dunnett post-test, comparing Squamous, Immunogenic, and ADEX to Progenitor). (**E**) Kaplan–Meier curves showing the overall survival of PDAC patients from the TCGA PAAD database, stratified according to *SRC* quartiles (low = bottom 25% (*n* = 37), and high = top 25% (*n* = 37)). A log-rank test was performed for survival analysis. (**F**) Gene sets enriched in the transcriptional profile of tumors belonging to the top *SRC* high-expression group, compared with the bottom expression group in the TCGA dataset series. Shown are the NES (normalized enrichment score) values for each pathway using the Hallmark genesets, meeting the following significance criteria: nominal *p*-value of <0.05, FDR <25%. Stem-related pathways are shown in red. (**G**) Representative enrichment plots for MTORC1 and NOTCH signaling. NES, *p*-value, and false discovery rate (FDR) q-values (q-val) are shown.

**Figure 2 ijms-21-07437-f002:**
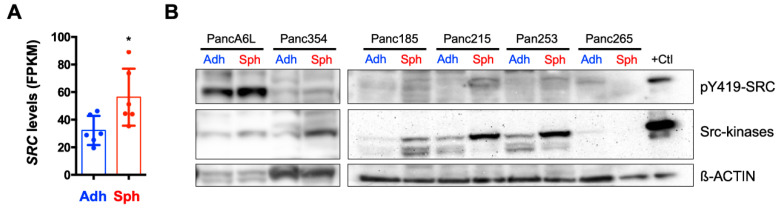
*SRC* is overexpressed in pancreatic cancer stem cells (PaCSCs). (**A**) Normalized Fragments Per Kilobase of transcript per Million mapped reads (FPKM) *SRC* levels from the published RNAseq analysis (ArrayExpress: E-MTAB-3808) of sphere and adherent cultures (PaCSCs and non-CSCs, respectively) derived from 3 different primary patient-derived xenograft (PDX) PDAC cultures (*n* = 2 biological duplicates per PDX culture). * *p* < 0.05; Student’s *t*-test. (**B**) Western blotting (WB) analysis of the expression of *SRC* kinases pY419–*SRC* protein expression in the six indicated PDX-derived samples cultured as adherent (Adh) monolayers or 3D spheres (Sph). ß-ACTIN was included as a loading control. A positive (+) *SRC*-overexpressing protein lysate control (+ Ctl) was included.

**Figure 3 ijms-21-07437-f003:**
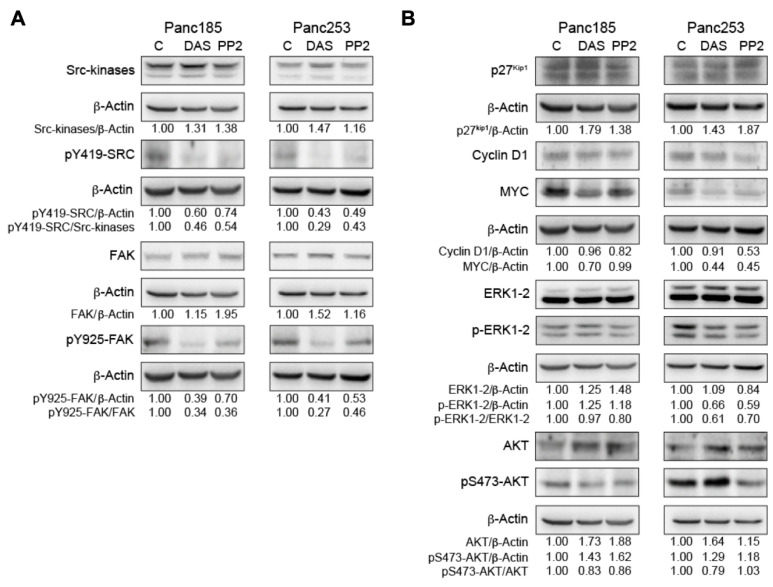
The effect of inhibition of *SRC* kinases on key signaling factors. (**A**) WB analysis of *SRC* kinases and pY419–*SRC* protein expression (top) or FAK and pY925–FAK (bottom) in control-, dasatinib (DAS)- or PP2-treated Panc185 or Panc253 PaCSCs. The indicated ratios for pY419–SRC/*SRC* kinases and pY925–FAK/FAK were determined, and the fold-changes are shown, setting diluent (C)-treated cells as 1.0. (**B**) WB analysis of p27^Kip1^, cyclin D1, MYC, ERK1-2, pERK1-2, AKT, or pS473–AKT protein levels in control-, DAS-, or PP2-treated Panc185 or Panc253 PaCSCs. The indicated ratios for p27^Kip1^/ß-actin, cyclin D1/ß-actin, MYC/ß-actin, pERK1-2/ERK1-2, and pS473–AKT/AKT were determined and the fold-changes are shown, setting diluent (C)-treated cells as 1.0. Densitometric analysis of the indicated bands was determined using ImageJ software, and all values were normalized to ß-actin values, which was included as a loading and normalization control.

**Figure 4 ijms-21-07437-f004:**
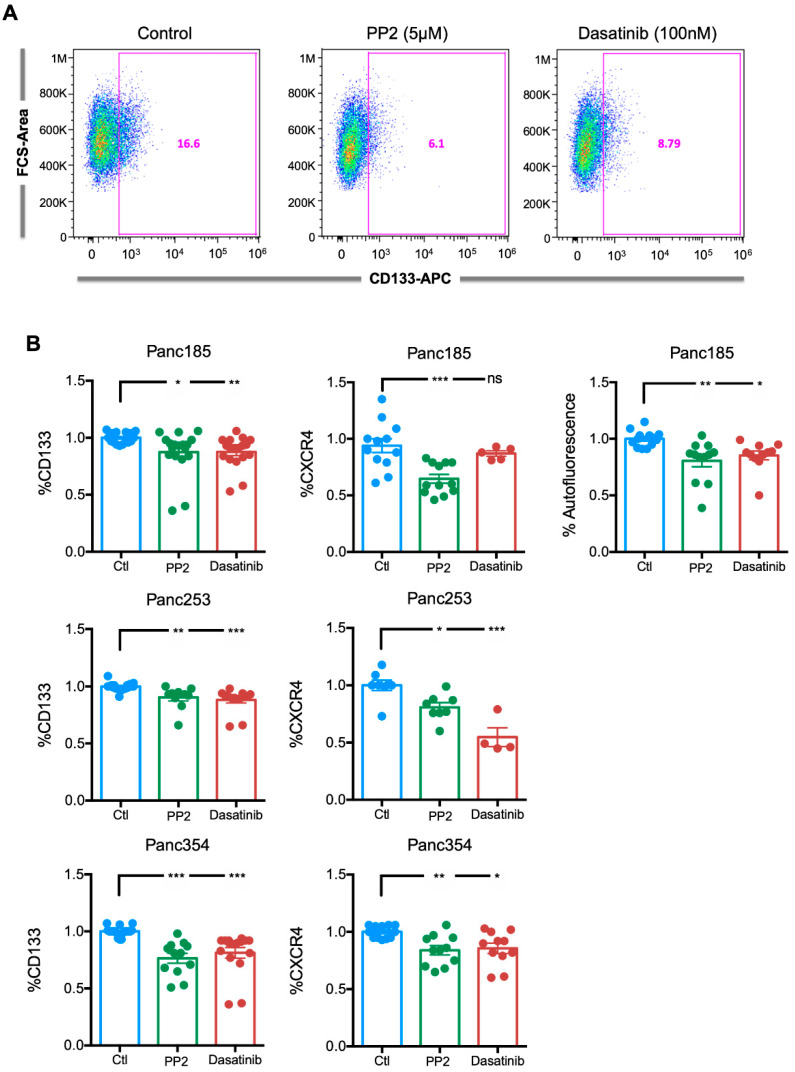
Inhibition of *SRC* kinases reduces the percentage of PaCSCs. (**A**) Representative flow cytometry plots for CD133–APC expression in control (diluent)-, PP2- or dasatinib-treated Panc185 cells. Percentages of CD133-APC-positive cells are shown. (**B**) Mean ± SEM of the fold change in the percentage of CD133-, CXCR4-, or autofluorescent-positive cells determined for control (diluent)-, PP2-, or dasatinib-treated Panc185, Panc253, or Panc354 cells. Diluent (Ctl)-treated cells were set as 1.0. Shown are the individual values calculated across a minimum of three independent experiments with at least *n* = 2–3 biological replicates. * *p* < 0.05, ** *p* < 0.01, ****p* < 0.001, ns = not significant, as determined by one-way ANOVA with Dunnett post-test, comparing PP2 and dasatinib to Ctl.

**Figure 5 ijms-21-07437-f005:**
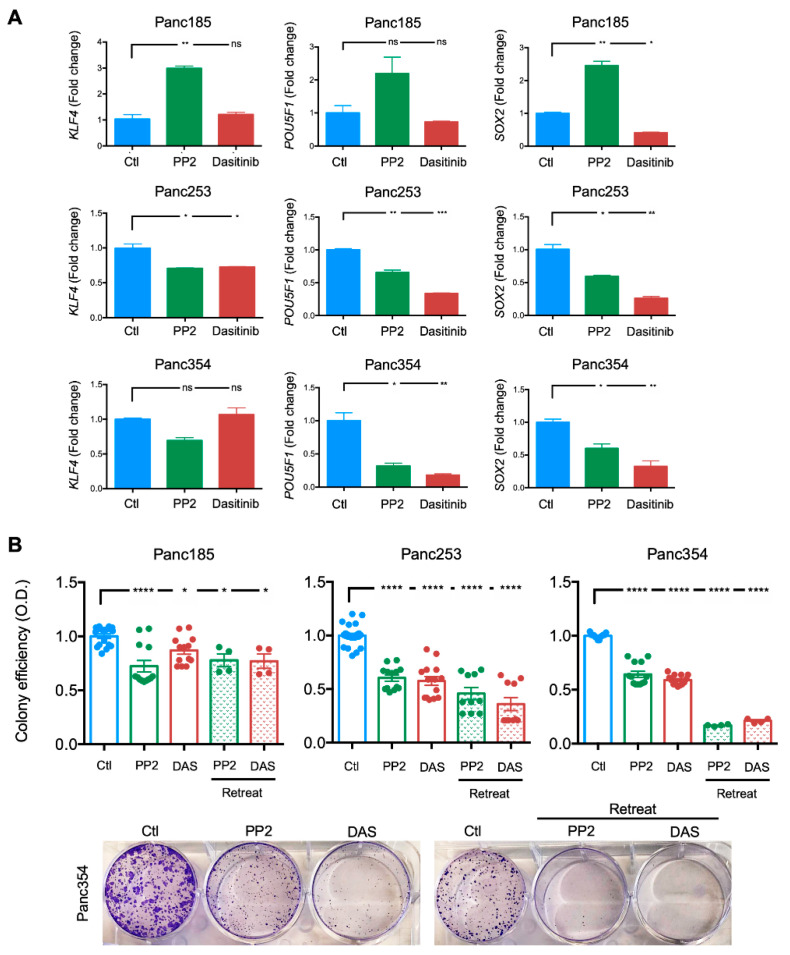
Inhibition of *SRC* kinases reduces the expression of pluripotency-associated genes and PaCSC colony formation efficiency. (**A**) Mean ± SD of the fold change in relative mRNA levels for *KLF4*, *POU5F1,* or *SOX2* determined in control (Ctl)-, PP2- or dasatinib-treated Panc185, Panc253, or Panc354 cells. Control (diluent)-treated cells were set as 1.0. * *p* < 0.05, ** *p* < 0.01, *** *p* < 0.001, ns = not significant; as determined by one-way ANOVA with Dunnett post-test, comparing PP2 and dasatinib to Ctl. (**B**) Mean ± SEM of the fold change in the percentage of colony formation efficiency for control-, PP2-, or dasatinib-treated Panc185, Panc253, or Panc354 cells (top). Control (diluent)-treated cells were set as 1.0. Retreat = at 7 days indicated treatments were readded. O.D. = optical density. Shown are the individual values calculated across a minimum of three independent experiments with at least *n* = 2–3 biological replicates. * *p* < 0.05, **** *p* < 0.0001, as determined by one-way ANOVA with Dunnett post-test, comparing treatments to Ctl. Representative images of crystal-violet-stained colonies from diluent (Ctl)-, PP2-, or dasatinib (DAS)-treated Panc354 cells 11 days post-seeding (bottom).

**Figure 6 ijms-21-07437-f006:**
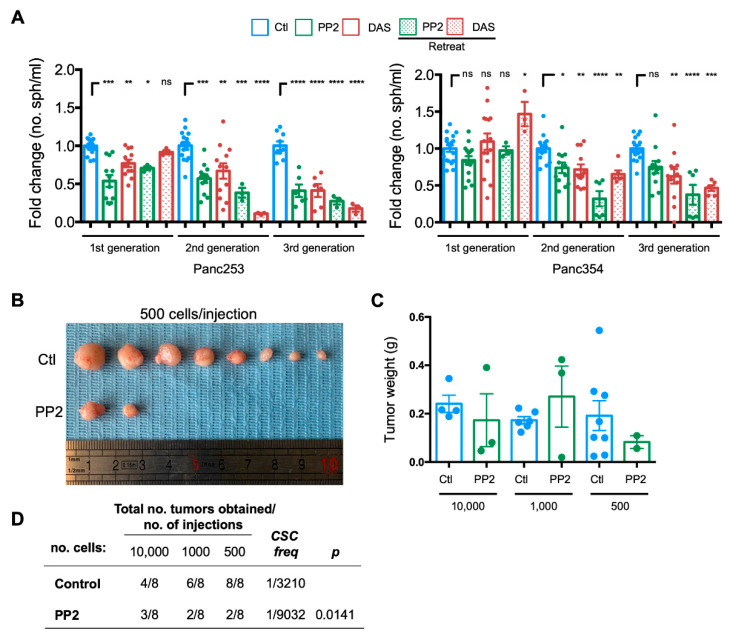
Inhibition of *SRC* kinases reduces PaCSC self-renewal and tumorigenesis. (**A**) Mean fold-change ± SEM of the number (no.) of first (1^st^), second (2^nd^), and third (3^rd^) generation spheres (sph)/mL determined for control-, PP2-, or dasatinib (DAS)-treated Panc253 or Panc354 cells. Diluent(Ctl)-treated cells were set as 1.0. Retreat = treatments were readded 5 days following the initiation of each generation. Shown are the individual values calculated across independent experiments, with at least *n* = 2–3 biological replicates. * *p* < 0.05, ** *p* < 0.01, *** *p* < 0.001, **** *p* < 0.0001, ns = not significant, as determined by one-way ANOVA with Dunnett post-test, comparing treatments to Ctl for each generation. (**B**) Representative images of tumors formed 10 weeks post-subcutaneous injection of 500 control diluent (Ctl)- or PP2-pretreated Panc354 PaCSCs. (**C**) Average tumor weights in grams (g) ± SD obtained in indicated groups. (**D**) Total number (no.) of tumors obtained/number of subcutaneous injections performed in the in vivo ELDA animal experiment (*n* = 4 mice per group) using indicated dilutions of control diluent- or PP2-pretreated Panc354 PaCSCs. Indicated CSC frequencies (freq) and *p*-values (*p*) were calculated using ELDA software.

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
