# Peer review of "Targeting SRC Kinase Signaling in Pancreatic Cancer Stem Cells"

_ijms, 2020, doi:10.3390/ijms21207437_

Round 1

Reviewer 1 Report

The manuscript by Alcalà and colleagues focuses on the role of the SRC TK in the pancreatic cancer stem cells behavior. The authors demonstrated that SRC inhibitors PP2 and DAS can be effective in blocking PaCSC growth both in vitro and in a preclinical model of pancreatic cancer.

The paper is very well written and the experiments are clearly reported. Even though the manuscript cannot be considered as completely original, it is remarkable the effort to demonstrate that SRC inhibitors can expand the currently limited repertoire of drugs that can be used in the treatment of PDAC, a tumor with dismal prognosis. No additional experiments are required by this reviewer.

Author Response

Authors’ response: We thank the reviewer for the positive evaluation of the study.

Reviewer 2 Report

This manuscript entitled "Targeting Src kinase signalling in pancreatic cancer stem cells", by Sonia Alcalà et al., highlights an important link between Src kinases and cancer stem cells properties in PDAC using PDX-derived preclinical models and pharmacological inhibitors. These findings might have a significant impact on the therapeutic strategies used for PDAC. The use of two various Src inhibitors (Dasatinib and PP2) is appropriate to attribute the observed effects to Src kinases targeting, since both of them can target other kinases at the concentration used. However, I have the Following comments and suggestions for the authors.

-Page 1 line 17: “of Src-family tyrosine kinases” is more appropriate than “of Src-family of tyrosine kinases”

-Page 4 Figure 1: The last panel is named “1E” instead of “1G”.

-Page 5 line 166: Fig.1G should be cited in the text.

-Page 8 Figure 4: To increase the visibility of the Figure 4B, I suggest to switch the graphs and the representative images (graphs on the top of the panel and images bellow) and modify the legend accordingly.

-Page 10 lines 313-317: Based on the images (Figure 6B), it seems that there is a significant difference in tumor size between Ctl and PP2 conditions, however, in the text it is noted that “although a notable but not-significant trend towards smaller tumors was observed …”. If all the obtained mice data (n=8, i.e. including the tumor volumes which were null) had been taken into consideration for the statistical analysis, the difference might have been really significant… Therefore, I suggest to remove the sentence “These data, might suggest that Src inhibition either inhibits tumor formation altogether, rather that slowing tumor incidence or growth”, in my opinion this is too speculative in view of the obtained results.

-Page 11 Figure 6: It seems there is a discrepancy in Figure 6 D, in the legend it’s mentioned n=4 mice per group and in the corresponding panel n=8.

Other general comments:

-The “Src2” annotation is confusing since it corresponds to the name of the anti-Src antibody clone used and SRC2/Src2 are respectively aliases for NCOA2/FGR genes. So, I recommend to replace “Src2” by “Src”, “Src kinases” or “SFKs” in all the western-blot panels (Figure 2, Figure 5 and Figure S3), the corresponding legends, and in the text.

-Note that the name of the anti-pSrc antibody (Table S1) has been recently change by the supplier (Invitrogen , ref #44-60G, phospho-Src Tyr419) since the position of the autophosphorylation site of the human Src sequence is not Y418 but Y419. The annotation of the WB pY416-Src in the Figure 5 corresponds to the chicken position. So, I recommend to change the annotation for the human position accordingly to the text and the rest of the manuscript.

-The addition of the activation level of phosphorylated Src kinases (pY419-Src WB) in Figure 2B could have been informative if available.

-Have the authors observed any correlation between SFK expression/activation level and spheres formation/number/size/sensitivity to Src inhibition? If available, these results could be incorporated and/or discussed in the discussion part of the manuscript to support the data.

Author Response

This manuscript entitled "Targeting Src kinase signalling in pancreatic cancer stem cells", by Sonia Alcalà et al., highlights an important link between Src kinases and cancer stem cells properties in PDAC using PDX-derived preclinical models and pharmacological inhibitors. These findings might have a significant impact on the therapeutic strategies used for PDAC. The use of two various Src inhibitors (Dasatinib and PP2) is appropriate to attribute the observed effects to Src kinases targeting, since both of them can target other kinases at the concentration used.

Authors’ response: We thank the reviewer for the critical evaluation of the study.

However, I have the Following comments and suggestions for the authors.

-Page 1 line 17: “of Src-family tyrosine kinases” is more appropriate than “of Src-family of tyrosine kinases”

Authors’ response: We have edited the text accordingly.

-Page 4 Figure 1: The last panel is named “1E” instead of “1G”.

Authors’ response: We thank the reviewer for catching this error. The panel has been renamed correctly.

-Page 5 line 166: Fig.1G should be cited in the text.

Authors’ response: We have edited the text accordingly.

-Page 8 Figure 4: To increase the visibility of the Figure 4B, I suggest to switch the graphs and the representative images (graphs on the top of the panel and images bellow) and modify the legend accordingly.

Authors’ response: We agree with the reviewer and have switched the graphs and the representative images as suggested.

-Page 10 lines 313-317: Based on the images (Figure 6B), it seems that there is a significant difference in tumor size between Ctl and PP2 conditions, however, in the text it is noted that “although a notable but not-significant trend towards smaller tumors was observed …”. If all the obtained mice data (n=8, i.e. including the tumor volumes which were null) had been taken into consideration for the statistical analysis, the difference might have been really significant… Therefore, I suggest to remove the sentence “These data, might suggest that Src inhibition either inhibits tumor formation altogether, rather that slowing tumor incidence or growth”, in my opinion this is too speculative in view of the obtained results.

Authors’ response: We have edited the section accordingly by removing the indicated sentence.

-Page 11 Figure 6: It seems there is a discrepancy in Figure 6 D, in the legend it’s mentioned n=4 mice per group and in the corresponding panel n=8.

Authors’ response: There is no discrepancy. Four mice were included per group, each mouse with two subcutaneous injections, yielding 8 injections in total per group. We have edited the label in panel 6D to avoid confusion.

Other general comments:

-The “Src2” annotation is confusing since it corresponds to the name of the anti-Src antibody clone used and SRC2/Src2 are respectively aliases for NCOA2/FGR genes. So, I recommend to replace “Src2” by “Src”, “Src kinases” or “SFKs” in all the western-blot panels (Figure 2, Figure 5 and Figure S3), the corresponding legends, and in the text.

Authors’ response: We have edited the figures as requested.

-Note that the name of the anti-pSrc antibody (Table S1) has been recently change by the supplier (Invitrogen , ref #44-60G, phospho-Src Tyr419) since the position of the autophosphorylation site of the human Src sequence is not Y418 but Y419. The annotation of the WB pY416-Src in the Figure 5 corresponds to the chicken position. So, I recommend to change the annotation for the human position accordingly to the text and the rest of the manuscript.

Authors’ response: We have edited the figures and text accordingly. We thank the reviewer for pointing out this recent change.

-The addition of the activation level of phosphorylated Src kinases (pY419-Src WB) in Figure 2B could have been informative if available.

Authors’ response: As requested we have reblotted the membranes to show basal levels of phosphorylated Src kinases (pY419-SRC) in Figure 2B.

-Have the authors observed any correlation between SFK expression/activation level and spheres formation/number/size/sensitivity to Src inhibition? If available, these results could be incorporated and/or discussed in the discussion part of the manuscript to support the data.

Authors’ response: We think this is a very interesting question; however, we have not observed any clear correlation and therefore would refrain from speculating whether SFK expression/activation level correlates with spheres formation, number, size or sensitivity to Src inhibition.

Reviewer 3 Report

In this article, the authors explored the role of the tyrosine kinase SRC in pancreatic cancer stem cells (PaCSCs). They found that SRC inhibitors, such as dasatinib and PP2, reduced the clonogenic, self-renewal, and tumor-initiating ability of PaCSCs. Moreover, at the molecular level, they observed changes in the expression of key members of the SRC pathway.

Although these are interesting findings, the following points should be addressed.

Major points:

  1. Although both PP2 and dasatinib are commonly defined as SRC family kinase inhibitors, they target many other kinases with similar potency (see, for instance, Bain et al., Biochem. J. 2007, 408, 297–315; Lombardo et al., J. Med. Chem. 2004, 47, 6658–6661; Bantscheff et al., Nat. Biotechnol. 2007, 25, 1035–1044). For this reason, to study specifically the role of SRC and other SRC family members in these cells, selective modulation of their expression (for instance through RNA interference) is necessary. Moreover, all the sentences ascribing the effects of PP2 and dasatinib exclusively to SRC inhibition should be changed throughout the text. Furthermore, the authors should clearly indicate that these drugs target also many other kinases and change the incorrect definition of PP2, which is not a selective SCR inhibitor (page 5, line 181).
  2. Section 2.2, Figure 2: In addition to SRC expression, the level of the active phosphorylated form of SRC should be compared between adherent and sphere cultures by the phospho-specific antibody.
  3. Section 2.3: the SRC activation status should be checked by using the phospho-specific antibody before and after treatment.
  4. Figure 5: The phosphorylated forms of SRC and FAK should be evaluated in the same membranes of their respective total forms.
  5. Statistical analyses should be revised. In particular, in Figure 1 (B, C, D), the Tukey’s multiple comparison test is used. However, although this test compares all pairs of data, the significance level is indicated only versus the first group in the plots (“Non-tumor” in B; “Adj. normal” in C; “Progenitor” in D). Thus, the Dunnett post-test, which compares all data versus a control, seems more appropriate. Moreover, the Tukey’s test is not mentioned in the “Statistical Analyses” sub-section (4.10). In Figures 3, 4, and 6, statistical analysis was performed by the Student’s t-test. However, this test is unsuitable for the multiple comparisons reported. One-way repeated measures ANOVA with Dunnett post-test seems more appropriate. In the “Statistical Analyses” sub-section (4.10), the sentence “one-way ANOVA (two-sided)” should be corrected because ANOVA does not have the “one-sided” and “two-sided” options. Moreover, in this section, the Bonferroni post-test is also mentioned, but this test is not mentioned in the figure legends. The sentence “Student’s t-test were used to determine differences between means of groups” should be changed to “Student’s t-test was used to determine differences between means of two groups”. The sentence “All analyses were performed using GraphPad Prism version 6.0 (San Diego California USA)” is repeated twice.

Minor points:

  1. The gene/protein names should be carefully revised throughout the text. In particular, according to the official nomenclature, symbols for both genes and proteins should be capitalized.
  2. Not all the abbreviations are explained at the first mention.
  3. Figure 1: the last panel should be G.
  4. Section 2.2: the last part of this section (starting from line 178, “to assess the biological….”) should be moved to section 2.3.
  5. For western blotting analyses (in both figures and text), SRC is indicated by the name of the antibody (Src2). However, this could be confounding.
  6. In the legends of Figure 3 and Figure 6, “Control-treated cells” should be changed to “untreated control cells”.
  7. Line 219: remove “the expression of”.
  8. Line 249 (“treatments were added on d0 and d5 post sphere formation”) and line 300 (“treatments were added on d0 or d0 and d5 post seeding for each generation”): please explain better when the treatments are performed following the seeding and when the effects are measured.
  9. Lines 280-281: please remove “less sensitive to Src kinase inhibition” and “more sensitive to Src kinase inhibition” because this is already reported above.
  10. Figure 5 B: since most of the changes in protein levels are very slight, the description of these effects should be toned-down.
  11. Lines 286-293 (“There is evidences……proliferation”): this part should be rewritten both for grammar mistakes and to improve clarity.

Author Response

In this article, the authors explored the role of the tyrosine kinase SRC in pancreatic cancer stem cells (PaCSCs). They found that SRC inhibitors, such as dasatinib and PP2, reduced the clonogenic, self-renewal, and tumor-initiating ability of PaCSCs. Moreover, at the molecular level, they observed changes in the expression of key members of the SRC pathway.

Authors response: First at all, we would like to thank the reviewer for his/her comments and criticisms, which have certainly helped us to improve the quality of the manuscript.

Although these are interesting findings, the following points should be addressed.

Major points:

  1. Although both PP2 and dasatinib are commonly defined as SRC family kinase inhibitors, they target many other kinases with similar potency (see, for instance, Bain et al., Biochem. J. 2007, 408, 297–315; Lombardo et al., J. Med. Chem. 2004, 47, 6658–6661; Bantscheff et al., Nat. Biotechnol. 2007, 25, 1035–1044). For this reason, to study specifically the role of SRC and other SRC family members in these cells, selective modulation of their expression (for instance through RNA interference) is necessary. Moreover, all the sentences ascribing the effects of PP2 and dasatinib exclusively to SRC inhibition should be changed throughout the text. Furthermore, the authors should clearly indicate that these drugs target also many other kinases and change the incorrect definition of PP2, which is not a selective SCR inhibitor (page 5, line 181).

Authors response: We completely agree with the reviewer, and are aware of all the references that you have provide us. Indeed, the expression we used - “selective inhibitors”- has created a misunderstanding What we pretended to say by writing “selective inhibitors”, is that PP2 and Dasatinib are not specific inhibitors, as both of them affect other kinases. Indeed, it was likely not the appropriate description to use, and as such we have eliminated the term “selective”. Nevertheless, the use of two different inhibitors, as pointed out by Reviewer 2, helps us to support our claim that Src-kinases are important for PaCSCs.

Transfection of an interfering RNA for c-Src would be very complicated as we are working with primary PDAC cells which are difficult to transfect (10-20% efficiency), limiting our ability to transiently modify c-Src levels in the CSC population. A pharmacological approach is more robust and uniform, and highlights the translational potential of our study.

  1. Section 2.2, Figure 2: In addition to SRC expression, the level of the active phosphorylated form of SRC should be compared between adherent and sphere cultures by the phospho-specific antibody.

Authors’ response: As requested we have reblotted the membranes to show basal levels of phosphorylated Src kinases (pY419-SRC) in Figure 2B.

  1. Section 2.3: the SRC activation status should be checked by using the phospho-specific antibody before and after treatment.

Authors’ response: We show in Figure 5 that the inhibitors of Src-kinases reduce Y419-SRC levels as would be expected. These WBs indicate the effectiveness of the inhibitors.  

  1. Figure 5: The phosphorylated forms of SRC and FAK should be evaluated in the same membranes of their respective total forms.

Authors’ response: In our laboratory stripping of western blots membranes does not always allow us to have a clean membrane at the end of the stripping process, causing problems in the second process of the membrane blotting. To avoid these possible problems, we decided to carry out the experiments in independent membranes, and to employ b-Actin as a house keeping protein to independently normalize expression for each of the ratio signals pY419-Src/b-Actin1 and Src-kinases/b-Actin2, as well as pY397-Fak/b-Actin3 and Fak/b-Actin4 ratio signals. We calculated the ratios: 1. pY419-Src/b-Actin1// Src-kinases/b-Actin2; 2. pY397-Fak/b-Actin3// Fak/b-Actin4; we then considered the values of Controls as 1 and determined the relative values for cellular treatments with Dasatinib (DAS) and with PP2. By doing so we believe that we can correctly evaluate the ratio pY419-Src/Src-kinases and pY397-Fak/Fak. We have previously used this methodology in other publications (PLoS One 2020, 15, e0235850; PLoS One 2017, e0188637; Oncotarget 2015, 6, 13520). Nevertheless, we carried out the WB with stripping for pY419-Src and pY397-Fak and then reblotted with Src-2 or Fak, the results, although they showed the same tendency, were of very poor quality. Thus, we maintained the original figure.

  1. Statistical analyses should be revised. In particular, in Figure 1 (B, C, D), the Tukey’s multiple comparison test is used. However, although this test compares all pairs of data, the significance level is indicated only versus the first group in the plots (“Non-tumor” in B; “Adj. normal” in C; “Progenitor” in D). Thus, the Dunnett post-test, which compares all data versus a control, seems more appropriate.

Authors’ response: We thank the reviewer for pointing this out, and we have reanalyzed the data using the Dunnett post-test, which has improved the significance of the data. The panels and corresponding legends have been updated.

Moreover, the Tukey’s test is not mentioned in the “Statistical Analyses” sub-section (4.10).

Authors’ response: We thank the reviewer for pointing this out, and the section has been updated accordingly.

In Figures 3, 4, and 6, statistical analysis was performed by the Student’s t-test. However, this test is unsuitable for the multiple comparisons reported. One-way repeated measures ANOVA with Dunnett post-test seems more appropriate.

Authors’ response: We thank the reviewer for pointing this out, and we have reanalyzed the data using One-way ANOVA with Dunnett post-test, which has improved the significance of the data. The panels and corresponding legends have been updated.

In the “Statistical Analyses” sub-section (4.10), the sentence “one-way ANOVA (two-sided)” should be corrected because ANOVA does not have the “one-sided” and “two-sided” options. Moreover, in this section, the Bonferroni post-test is also mentioned, but this test is not mentioned in the figure legends. The sentence “Student’s t-test were used to determine differences between means of groups” should be changed to “Student’s t-test was used to determine differences between means of two groups”. The sentence “All analyses were performed using GraphPad Prism version 6.0 (San Diego California USA)” is repeated twice.

Authors’ response: We sincerely thank the reviewer for the careful analysis of the “Statistical Analyses” sub-section (4.10), which has now been updated according to his/her recommendations.

Minor points:

  1. The gene/protein names should be carefully revised throughout the text. In particular, according to the official nomenclature, symbols for both genes and proteins should be capitalized.

Authors’ response: We have carefully revised the manuscript to ensure that symbols for both genes and proteins are capitalized and italized (for mRNAs).

  1. Not all the abbreviations are explained at the first mention.

Authors’ response: We apologize for this oversight and have made sure that all abbreviations are defined at first mention.

  1. Figure 1: the last panel should be G.

Authors’ response: We thank the reviewer for catching this error. The panel has been renamed correctly.

  1. Section 2.2: the last part of this section (starting from line 178, “to assess the biological….”) should be moved to section 2.3.

Authors’ response: We have moved the text as requested.

  1. For western blotting analyses (in both figures and text), SRC is indicated by the name of the antibody (Src2). However, this could be confounding.

Authors’ response: We agree, and as requested by Reviewer no. 2’s we have replaced Scr2 with Src-kinases, as the Src2 antibody recognizes c-Src, Yes and Fyn.

  1. In the legends of Figure 3 and Figure 6, “Control-treated cells” should be changed to “untreated control cells”.

Authors’ response: We have changed control-treated cells to diluent-treated cells as these cells were treated with a similar concentration of DMSO, the diluent used to resuspend the inhibitors.

  1. Line 219: remove “the expression of”.

Authors’ response: As requested, “the expression of” has been removed.

  1. Line 249 (“treatments were added on d0 and d5 post sphere formation”) and line 300 (“treatments were added on d0 or d0 and d5 post seeding for each generation”): please explain better when the treatments are performed following the seeding and when the effects are measured.

Authors’ response: We apologize for this confusion and have re-written the text corresponding to re-treatment approach used in the sphere formation assays (now lines 310-314)

  1. Lines 280-281: please remove “less sensitive to Src kinase inhibition” and “more sensitive to Src kinase inhibition” because this is already reported above.

Authors’ response: This has been removed as requested.

  1. Figure 5 B: since most of the changes in protein levels are very slight, the description of these effects should be toned-down.

Authors’ response: We agree with the reviewer and we have changed the description of the results for Figure 5B. We refer the reviewer to lines: 300-306.

  1. Lines 286-293 (“There is evidences……proliferation”): this part should be rewritten both for grammar mistakes and to improve clarity.

Authors’ response: Once again, we appreciate the reviewer’s careful eye. Indeed, this paragraph had grammar mistakes and it was a bit confusing. We have re-written the section and corrected these mistakes and improved its overall understanding.

Round 2

Reviewer 3 Report

The authors addressed many criticisms and adequately explained why they could not meet some of the reviewer’s requests. However, there are still some minor points to be resolved.

  1. The issue of the target selectivity of SRC kinase inhibitors is insufficiently considered. Although the authors expressed agreement with the reviewer’s comment, they just removed the term “selective”. The reviewer understands the difficulties in transfecting primary PDAC cells with siRNAs, which can inhibit selectively SRC. However, the authors should at least state in section 2.3 that dasatinib and PP2 target also other kinases and provide opportune references. In this section, they, instead, described the two drugs just as inhibitors of SFKs (in a very repetitive manner: “two different inhibitors of Src tyrosine kinase activity, Dasatinib (DAS) and PP2”; “Pyrazolopyrimidine compound PP2 is an inhibitor of SFKs”; “DAS (BMS-354825, Sprycel) is a potent small molecule inhibitor of Src-kinases”). Moreover, the authors should briefly discuss in the “discussion” section that these drugs target also many other kinases, the inhibition of which can, therefore, contribute to the effects observed.

  1. Many protein symbols have now been capitalized, according to the official nomenclature; however, the symbols for the SRC protein and other members of the SRC family are still in lowercase letters. Moreover, SRC mRNA (which is the correct form) is now indicated as c-Src (SRC), which is confusing.

Author Response

We thank Reviewer no3 for the opportunity to revise and improve the submitted manuscript. We have provided our responses to the 2 points raised.

  1. The issue of the target selectivity of SRC kinase inhibitors is insufficiently considered. Although the authors expressed agreement with the reviewer’s comment, they just removed the term “selective”. The reviewer understands the difficulties in transfecting primary PDAC cells with siRNAs, which can inhibit selectively SRC. However, the authors should at least state in section 2.3 that dasatinib and PP2 target also other kinases and provide opportune references. In this section, they, instead, described the two drugs just as inhibitors of SFKs (in a very repetitive manner: “two different inhibitors of Src tyrosine kinase activity, Dasatinib (DAS) and PP2”; “Pyrazolopyrimidine compound PP2 is an inhibitor of SFKs”; “DAS (BMS-354825, Sprycel) is a potent small molecule inhibitor of Src-kinases”). Moreover, the authors should briefly discuss in the “discussion” section that these drugs target also many other kinases, the inhibition of which can, therefore, contribute to the effects observed.

Author's response: We apologize that we did not sufficiently and adequately address this point the the last revision. We completely agree with the issue raised by the reviewer. As such, in section 2.3. (lines 198-201), we have rewritten the indicated text as such "Pyrazolopyrimidine compound PP2 is primarily known as an inhibitor of SFKs [59], but PP2 can also inhibit other kinases, such as kinase activity of the TGF-β type I receptor activin receptor-like kinase 5 [60]. DAS (BMS-354825, Sprycel) is a potent small molecule that inhibits greater than 40 different protein kinases, including SFKs and other non-RTKs such as FRK, BRK, and ACK [61].
Likewise, and as suggested, we have briefly discussed in the “discussion” section that these drugs also target many other kinases. Specifically lines 437-442, we have added "It is important to stress that PP2 and DAS are broad-specificity tyrosine kinase inhibitors, as mentioned above. Thus, while both compounds can target and reduce the activity of SFKs, as demonstrated herein, their effects may reach beyond that of SFKs. Therefore, we cannot rule out the possibility that the biological and functional effects observed with PP2 and DAS may be due to SFK-independent affects. As such, genetic based approaches to silence or knockout SRC and/or other SKFs specifically in PaCSCs are warranted."
We hope that these changes and additions are satisfactory.

Finally, and equally important, we have changed the order of the figures, placing the protein signalling analyses before the PaCSC phenotypic and functional characterization assays. We feel that it is important to show early on that DAS and PP2 do reduce pY419-SRC/SRC-kinases levels.

  1. Many protein symbols have now been capitalized, according to the official nomenclature; however, the symbols for the SRC protein and other members of the SRC family are still in lowercase letters. Moreover, SRC mRNA (which is the correct form) is now indicated as c-Src (SRC), which is confusing.

Author's response: We apologize again and agree with the Reviewer. Rest assured we now refer to SRC throughout as SRC when referring to mRNA levels or SRC when referring to protein levels or to SRC in general. Likewise, all other proteins names mentioned have been capitalized.

We sincerely thank the reviewer for the opportunity to improve the study and we hope that the changes made now resolve any issues or concerns.